# Integrins Increase Sarcoplasmic Reticulum Activity for Excitation—Contraction Coupling in Human Stem Cell-Derived Cardiomyocytes

**DOI:** 10.3390/ijms231810940

**Published:** 2022-09-19

**Authors:** Brian X. Wang, Christopher Kane, Laura Nicastro, Oisín King, Worrapong Kit-Anan, Barrett Downing, Graziano Deidda, Liam S. Couch, Christian Pinali, Anna Mitraki, Kenneth T. MacLeod, Cesare M. Terracciano

**Affiliations:** 1National Heart & Lung Institute, Imperial College London, London SW7 2AZ, UK; 2Department of Metabolism, Digestion and Reproduction, Imperial College London, London SW7 2AZ, UK; 3Human Safety, Bayer Crop Science, 06903 Sophia-Antipolis, France; 4Institute of Electronic Structure and Laser (IESL), Foundation for Research and Technology−Hellas (FORTH), 700 13 Heraklion, Greece; 5Department of Materials Science and Technology, University of Crete, 700 13 Heraklion, Greece; 6Division of Cardiovascular Sciences, University of Manchester, Manchester M13 9NT, UK; 7Laboratory of Myocardial Electrophysiology, 4th Floor, Imperial Centre for Translational and Experimental Medicine, Imperial College London, Du Cane Road, London W12 0NN, UK

**Keywords:** stem cell, integrin, Ca^2+^ cycling, cardiomyocyte, fibrosis, sarcoplasmic reticulum, ryanodine receptor

## Abstract

Engagement of the sarcoplasmic reticulum (SR) Ca^2+^ stores for excitation–contraction (EC)-coupling is a fundamental feature of cardiac muscle cells. Extracellular matrix (ECM) proteins that form the extracellular scaffolding supporting cardiac contractile activity are thought to play an integral role in the modulation of EC-coupling. At baseline, human induced pluripotent stem cell-derived cardiomyocytes (hiPSC-CMs) show poor utilisation of SR Ca^2+^ stores, leading to inefficient EC-coupling, like developing or human CMs in cardiac diseases such as heart failure. We hypothesised that integrin ligand–receptor interactions between ECM proteins and CMs recruit the SR to Ca^2+^ cycling during EC-coupling. hiPSC-CM monolayers were cultured on fibronectin-coated glass before 24 h treatment with fibril-forming peptides containing the integrin-binding tripeptide sequence arginine–glycine–aspartic acid (2 mM). Micropipette application of 40 mM caffeine in standard or Na^+^/Ca^2+^-free Tyrode’s solutions was used to assess the Ca^2+^ removal mechanisms. Microelectrode recordings were conducted to analyse action potentials in current-clamp. Confocal images of labelled hiPSC-CMs were analysed to investigate hiPSC-CM morphology and ultrastructural arrangements in Ca^2+^ release units. This study demonstrates that peptides containing the integrin-binding sequence arginine–glycine–aspartic acid (1) abbreviate hiPSC-CM Ca^2+^ transient and action potential duration, (2) increase co-localisation between L-type Ca^2+^ channels and ryanodine receptors involved in EC-coupling, and (3) increase the rate of SR-mediated Ca^2+^ cycling. We conclude that integrin-binding peptides induce recruitment of the SR for Ca^2+^ cycling in EC-coupling through functional and structural improvements and demonstrate the importance of the ECM in modulating cardiomyocyte function in physiology.

## 1. Introduction

Human cardiomyocytes (CMs) are characterised by a finely tuned ultrastructure sensitive to external dynamic factors, in a phenomenon known as myocardial plasticity [1,2]. Use of human induced pluripotent stem cell-derived CMs (hiPSC-CMs) has been hindered by the failure of these cells to reproducibly develop features of the mature healthy CM in vitro [3,4]. Despite their immaturity in vitro, hiPSC-CMs transplanted into the heart undergo differentiation (at least partially) into the required phenotype [5], demonstrating their suitability for studying myocardial plasticity.

Regulation of cytoplasmic Ca^2+^ is a fundamental component of excitation–contraction (EC) coupling. To adapt to the increased postnatal and adult requirements, immature CMs engage the sarcoplasmic reticulum (SR) as the primary regulator of Ca^2+^ cycling [6]. In stem cell derived-CMs, Ca^2+^ cycling is proportionally more dependent on sarcolemmal influx compared to release of Ca^2+^ from SR stores [7]. Our group has recently shown that human fibroblast-secreted exosomes, vesicles containing genetic material and extracellular matrix (ECM) proteins, abbreviate Ca^2+^ cycling, bring the electrophysiological features of hiPSC-CMs closer to that of the physiological, adult CM [8]. However, the modulators of this increase in Ca^2+^ cycling efficiency and SR recruitment are largely unknown.

Human CMs respond to changes in the ECM via integrins; heterodimeric transmembrane receptors that function mechanically by connecting the cytoskeleton to the ECM and biochemically by activating intracellular signalling cascades [9]. The integrin-binding domain in ECM proteins is the arginine–glycine–aspartic acid (RGD) tripeptide. RGD-containing peptides can be used to investigate the role of integrins in the functional recruitment of the SR—the availability of Ca^2+^ stores for EC-coupling.

We show that integrin ligands modulate EC-coupling through functional regulation of SR and structural regulation of SR-sarcolemma junctions. The integrin ligand–receptor interactions provide a mechanism for transduction of mechano- and biochemical reception into dynamic changes in CM contractility. Biocompatible integrin ligands present a novel mechanism for modulating hiPSC-CM maturation and CM function in disease.

## 2. Results

### 2.1. Modulation of Ca^2+^ Cycling by Soluble Integrin Ligand

To gain insight into the regulation of Ca^2+^ cycling by integrin ligands, preliminary studies cultured CMs on glass dishes with various combinations of ECM proteins (Matrigel, fibronectin, MapTrix hygel) for 24 h before optical recording at 1 Hz field-stimulation, identifying no change in Ca^2+^ cycling parameters (Appendix A). Soluble peptides containing the RGD tripeptide motif have previously been shown to activate cytoskeletal assembly of integrin-associated signalling proteins in CMs, as well as mirroring in vivo downstream pathways under conditions of increased mechanical load [8,9]. We incubated confluent hiPSC-CM monolayers with soluble integrin ligands for 24 h before optical recordings at 1 Hz field-stimulation (Appendix A). hiPSC-CMs incubated with a range of RGD-containing peptides demonstrated that the flanking sequences on either side of the integrin-binding tripeptide determines the potency of Ca^2+^ transient modulation. Negative control RGES (Arg-Gly-Glu-Ser) had no effect on Ca^2+^ transient parameters. Soluble RGDS and GRGDSP shortened TTP (Appendix A; *p* = 0.0263 and *p* = 0.0002, respectively) with no change in T80 (Appendix A), demonstrating that there are separate mechanisms mediating an increase in cytosolic Ca^2+^ availability and cytosolic Ca^2+^ removal. 

The notion that flanking sequences determine RGD potency in abbreviating Ca^2+^ cycling was further strengthened by experiments in which we incubated hiPSC-CMs with soluble GRGDS (Gly-Arg-Gly-Asp-Ser), (Appendix A). More specifically, GRGDS caused a reduction in Ca^2+^ transient time to peak (TTP) (Appendix A; *p* = 0.0016) and time from peak to 80% back to baseline (T80) (Appendix A; *p* = 0.0039), indicating a role in regulating both Ca^2+^ release into the cytoplasm and Ca^2+^ removal, required for the initiation and cessation of CM contraction, respectively. Collagen, a common ECM protein, did not mediate any changes in Ca^2+^ transient morphology (Appendix A), indicating that the potency of the RGD motif is determined by availability of the motif for integrin binding.

A confounding factor to the use of soluble GRGDS was that, while the hiPSC-CMs were initially cultured as a confluent monolayer (Appendix A), GRGDS caused cell detachment from the glass dishes. This phenomenon was observed for both GRGDS in solution and the collagen-GRGDS gel (Appendix A, respectively). Myosin II inhibitor, blebbistatin, acts as an EC-uncoupler. Pre-treatment of GRGDS-treated hiPSC-CMs with blebbistatin prevented cell detachment, although morphological changes were observed, with an apparent reduction in cell surface area forming a stellate pattern (Appendix A), in contrast to the confluent monolayer in control (Appendix A). Blebbistatin only partially inhibited the GRGDS-induced reduction in TTP (Appendix A; *p* < 0.0001), with no change in GRGDS-mediated abbreviation in T80 (Appendix A; *p* = 0.601), indicating that GRGDS-induced abbreviation of Ca^2+^ transients is independent of cell detachment. 

### 2.2. Modulation of Ca^2+^ Cycling by Fibril-Forming Integrin Ligand 

The detachment of GRGDS-treated hiPSC-CMs from the substrate prevented us from attributing GRGDS-induced changes in signal amplitude of our non-ratiometric Ca^2+^ sensitive dye to changes in Ca^2+^ availability [10,11,12,13]. To overcome this confounding factor, we investigated the effects of a bifunctional self-assembling amyloid peptide based on the β-amyloid-forming sequence of the adenovirus fibre shaft [14]. The fibril-forming amino acid sequence of RGDSGAITIGC (ff_RGD) did not reduce cell viability (Appendix A; *p* = 0.691) and, unlike GRGDS, did not cause cell detachment from the glass substrate (Appendix A; *p* = 0.484), indicating that it is a suitable alternative to soluble integrin ligands for investigating integrin ligand–receptor interactions.

Optical recordings of ff_RGD-treated hiPSC-CMs field-stimulated at 1 Hz showed that ff_RGD, like GRGDS following pretreatment with blebbistatin, abbreviated the Ca^2+^ transients (Figure 1A) by reducing TTP (Figure 1B; *p* = 0.0016) and T80 (Figure 1C; *p* = 0.0055), in the absence of changes in Ca^2+^ transient amplitude. The weaker efficacy of ff_RGD compared to GRGDS may be, at least in part, accounted for by the differences in availability of RGD—in ff_RGD, the integrin-binding RGD tripeptide is likely to have a lower exposure to hiPSC-CMs than in the soluble GRGDS solution.

The lack of cell detachment following ff_RGD treatment afforded us the opportunity to analyse microelectrode recordings of APs in current-clamp in control and ff_RGD-treated hiPSC-CM monolayers (Figure 1D). An increase in spontaneous beating of ff_RGD-treated hiPSC-CM monolayers (Figure 1E; *p* = 0.0018) was accompanied by an abbreviation in APD (Figure 1F; *p* = 0.0029). There was no change in the maximum diastolic membrane potential (Figure 1G; *p* = 0.602) or AP amplitude.

### 2.3. ff_RGD-Mediated Engagement of the SR for Ca^2+^ Cycling

The ability of ff_RGD to abbreviate Ca^2+^ transients in hiPSC-CM monolayers without changing the cell density afforded us the ability to assess the contributions of Ca^2+^ transporters to Ca^2+^ extrusion [15]. Locally applied caffeine induced an increase in cytoplasmic Ca^2+^ in both the standard and Na^+^/Ca^2+^-free Tyrode’s solution (Figure 2A,B). In keeping with a reduced T80 (Figure 1C), ff_RGD increased Ca^2+^ transient decay rate (Figure 2C; *p* = 0.0446), attributed to a 63% increase in SR Ca^2+^ uptake rate (Figure 2D; *p* = 0.0049). Ff_RGD had no effect on Na^+^−Ca^2+^ exchanger (NCX) (Figure 2E; *p* = 0.941) or the slow Ca^2+^ removal mechanisms (Figure 2F; *p* = 0.856). The SR:NCX ratio is an important indicator of Ca^2+^ cycling maturity [16]. The SR:NCX ratio changed from 49:51 in control to 62:38 with ff_RGD (Figure 2G). Ff_RGD did not change the SR content (Figure 2H; *p* = 0.376) but increased SR fractional release (Figure 2I; *p* = 0.0426). These results demonstrate that ff_RGD increases SR contribution to Ca^2+^-induced Ca^2+^ release (CICR) and Ca^2+^ removal required for cessation of contractile activity.

### 2.4. Integrin-Mediated Regulation of Cell Morphology and Sarcomere Length

Following 24 h treatment with ff_RGD, we observed key changes in hiPSC-CM morphology (Figure 3A,B). Notably, we identified an increase in cell aspect ratio (Figure 3C; *p* = 0.0021), closer to that of the adult human CM. By labelling α-actinin (Figure 3D,E), we identified an increase in sarcomere length following ff_RGD treatment vs. control (Figure 3F; 1.75 ± 0.0309 μm vs. 1.63 ± 0.0201 μm, *p* = 0.0039), a key determinant of contractile activity. Other key changes include a reduction in hiPSC-CM cross-sectional area and volume (Appendix A). Adult human CMs are significantly larger than hiPSC-CMs. Rather than acting in isolation, further biochemical or mechanical stimuli beyond integrin ligand–receptor activity are likely to drive the increase in cell area and volume in the developing CM.

### 2.5. Integrin-Mediated Co-Localisation of RyRs and LTCCs

SR Ca^2+^ release occurs at the junction between the SR domains closest to the sarcolemma (junctional SR (jSR)) that bears the Ca^2+^ release channels (ryanodine receptor (RyR2)) and the sarcolemmal L-type voltage-gated Ca^2+^ channels (LTCCs). Following immunofluorescence labelling of the 1.2α subunit of LTCCs and RyR2 in control (Figure 4A,B) and ff_RGD-treated (Figure 4C,D) monolayers, Mander’s coefficient of LTCCs overlapped with the background of RyR pixels (M1) increased following ff_RGD treatment (Figure 4E; *p* = 0.0363). There was no difference in the Mander’s coefficient of RyR2 overlapped with a background of LTCC pixels (M2) (Figure 4F; *p* = 0.1143). Pearson’s correlation coefficient between LTCC and RyR2 increased following ff_RGD treatment (Figure 4G; *p* = 0.001). These findings indicate that integrin ligand–receptor interactions mediate co-localisation of RyR2 with sarcolemmal LTCCs responsible for CICR, in keeping with abbreviation in Ca^2+^ transient TTP (Figure 1B).

### 2.6. Integrin-Mediated Regulation of SR-Sarcolemma Junctions 

The junctions between the jSR and sarcolemma function as Ca^2+^ release units (CRUs)—the sites at which Ca^2+^ sparks are detected [17]. In adult CMs, these junctions are 12–15 nm wide to host the macromolecular complexes containing RyR2 and LTCCs situated in juxtaposition. Electron microscopy of hiPSC-CMs (Appendix A) showed that ff_RGD increased the junctional gap distance, closer to the physiological range of 2–2.2 μm (Appendix A; *p* = 0.0002).

## 3. Discussion

Our data demonstrate that integrin ligands abbreviate the poorly orchestrated CICR in hiPSC-CMs, increasing the rate of Ca^2+^ rise, as well as increasing the rate of Ca^2+^ decay by increasing the contribution of the SR to Ca^2+^ cycling. The EC-coupling changes are largely mediated by integrin-induced actin polymerisation. Fibril-forming integrin ligand ff_RGD induce changes in CM ultrastructure, including an increased co-localisation between LTCCs and RyR2, key proteins in CM Ca^2+^ cycling.

### 3.1. Integrin-Mediated ECM Cell Adhesions and Signalling

Integrins are cell surface receptors integral to the processes of cellular adhesion, mechanosensing and signalling. Mammals have been found to express over 18 α and 8 β integrin subunits, which heterodimerise to form 24 receptors [18]. Of these, 8 have been identified to bind to RGD-containing peptides [19]. A major site of integrin localization is at costameres, the site at which Z-lines of the myofibrils anchor to the sarcolemma [20]. Integrins are also found at intercalated discs (ICDs), which connect CMs end-to-end [21]. These sites contain structures integral to both mechanical and electrical coupling. Integrin expression at costameres has previously been shown to be regulated by mechanical force —there is loss of integrin expression when contraction is arrested, and an increase in expression when CMs are stretched [22]. Integrins are therefore well positioned to modulate CM calcium cycling physiologically and in pathology, yet their role remains largely unexplored. 

It has been reported that integrin-mediated regulation of cell function is through connections to the actin cytoskeleton and signalling via recruitment of focal adhesion molecules to form the focal adhesion complex (FAC) [23,24,25]. Ligand binding to the extracellular domain of integrins induces the cytoplasmic portion of receptor subunits to form interactions with cytoskeleton proteins, which then form focal adhesions to ECM components [26]. Importantly, only RGD-containing materials peptides within a three-dimensional collagen gel can induce FAC formation, but integrins bound to RGD-containing peptides in solution are unable to cluster and recruit focal adhesion proteins onto the cytoskeletal complex [23,24]. Leveraging on this difference, we treated hiPSC-CMs with collagen gel alone, collagen-GRGDS or GRGDS alone (Appendix A). hiPSC-CMs did not display any changes in Ca^2+^ transient properties when the collagen gel was present. Also, there was no change in the GRGDS-mediated abbreviation of the Ca^2+^ transient when it was applied in a collagen gel. This indicates that the abbreviation of Ca^2+^ transients by soluble RGD-containing compounds is independent of FAC formation.

### 3.2. Ff_RGD as a Fibril-Forming Integrin Ligand

The current study used the fibril-forming amino acid sequence of RGDSGAITIGC, referred to as ff_RGD, to mimic fibronectin. The amino acid sequence of ff_RGD was built around the naturally occurring sequence NSGAITIG observed in the adenovirus fibre shaft at residues 385–392, identified to be the key amyloidogenic region of NSGAITIGC that led to self-assembly into amyloid fibril peptide nanostructures [27,28]. Studies identified that the NS residues could undergo substitution without affecting the assembling properties of the amyloidogenic motif GAITIG [29]. Experimental and computational studies showed that ff_RGD preserves the β-sheet core GAITIG of the NSGAITIG self-assembling sequence and simultaneously displays the integrin-binding RGD sequence at terminal peptide positions for integrin exposure [14]. Serine (S) is a flexible residue that links the RGD motif and the GAITIG motif. This allows the sequence to favour a spatial arrangement like the tenth type III repeat of fibronectin, characterised by an exposed RGD-sequence in a loop configuration [30]. Molecular dynamic simulation studies have shown that the architecture of the ff_RGD sequence of RGDSGAITIGC post-assembly appears as 4-stranded antiparallel β-sheets, where the integrin-binding RGD motif and cysteine are externally exposed from the fibril core formed by the building block GAITIG amino acid sequence [14]. Moreover, the same study identified that geometrical analysis of solvent exposure of the RGD motif in fibronectin is comparable to the solvent exposure of the RGD motif in the highly ordered, antiparallel ff_RGD fibrils. Consequently, the exposed RGD tripeptides are available for binding integrins and thus fulfil adhesion function. 

### 3.3. Engagement of the SR Ca^2+^ Stores for Ca^2+^ Cycling

There are two main differences when comparing the EC-coupling kinetics of healthy adult human CMs to hiPSC-CMs: hiPSC-CMs display slower rise in reaching Ca^2+^ transient peak amplitude and slower SERCA and NCX Ca^2+^ transport [15,16]. In hiPSC-CMs, the rise in Ca^2+^ availability is due to an initial rise in Ca^2+^ at the sarcolemma, which spreads inward [31], an inefficient form of CICR. Integrin ligands abbreviated the Ca^2+^ transient TTP without changing the amplitude, allowing us to rule out the changes in Ca^2+^ availability during peak contraction as the trigger for changes in kinetics.

The Ca^2+^ transient amplitude, an indicator of contractile force, is largely determined by stretch force exerted on CMs [32]. hiPSC-CM force generation is 15 nN/cell [33], much lower than the 12.6 μN reported for single adult ventricular myocytes [34]. There was no significant change in amplitude following integrin ligand treatment. The mechanical force in the contracting myocardium is likely to play a large role in determining force generation and the integrin-mediated changes in subcellular organisation play a permissive role, enabling developing CM or hiPSC-CMs transplanted into the heart to respond to mechanical stimuli. 

The ratio of SR uptake to NCX activity is often used as a surrogate marker for the maturity of CM Ca^2+^ cycling [15]. In healthy human adult ventricular CMs, this ratio is approximately 77:23 and reduces to 64:36 in failing CMs [35]. Ff_RGD changed the SR:NCX ratio from 49:51 in control conditions to 62:38 following incubation with integrin ligand ff_RGD (Figure 2G). The increase in SR-mediated Ca^2+^ removal from the cytoplasm and SR fractional release in the absence of a concomitant change in SR Ca^2+^ content indicates that RyR2 is more responsive to Ca^2+^ with a shorter opening time. The shift in SR Ca^2+^ uptake dependence mirrors our previous observations that co-culturing human CMs with human cardiac fibroblasts, the main modulators of ECM turnover, induces more efficient CM Ca^2+^ cycling compared to CM monocultures [13].

Alongside these improvements in Ca^2+^ cycling, we demonstrate increases in spontaneous beating rate. hiPSC-CMs have previously been observed to exhibit a negative force–frequency relationship when cultured in monolayer [36], thought to possibly result from factors associated with their immaturity, such as a random sarcomere alignment, lack of transverse tubule organisation, and their undeveloped Ca^2+^ handling mechanisms [37]. Since we observed improved efficiency of CICR at paced frequencies, the effect on beating rate is likely a consequence of improvement in overall Ca^2+^ cycling machinery and CM ultrastructure induced by integrin binding motifs within the ECM, as demonstrated in this study. 

### 3.4. Regulation of hiPSC-CM AP Morphology 

APD in adult CMs varies depending on the subtype but lies between 100–300 ms [38]. Our study demonstrated that treatment of spontaneously beating hiPSC-CMs with ff_RGD causes an abbreviation of APD and an increase in beating rate. Typically, CMs demonstrate APD restitution, whereby as diastolic interval decreases the APD at late repolarisation also decreases. CM AP restitution is often corrected using Bazett’s formula. Our previous study identified that hiPSC-CMs show limited, if any, APD restitution when using Bazett’s formula [39]. Therefore, rate correction of APD is not appropriate in our current study. 

This study demonstrates that hiPSC-CMs treated with integrin ligands are less reliant on NCX-mediated Ca^2+^ extrusion. This reduces the inward current and is expected to shorten the APD. However, the APD is determined by a myriad of other transporters, and this reduced inward current produced by NCX could be compensated. Future studies should delineate the relationship between AP morphology and Ca^2+^ cycling in hiPSC-CMs.

Although rapid stimulation of hiPSC-CMs has been shown to drive maturation [40], we show that co-culture with RGD-containing peptides is an alternative to electrochemical stimulation for driving the maturation of hiPSC-CMs. The ability of hiPSC-CMs to synchronise with the native myocardium is a key determinant in the success of implantation. Therefore, integrin ligand–receptor interactions between ECM proteins and hiPSC-CMs are likely important in determining the success or failure of implanted hiPSC-CMs synchronising with and incorporating into the native myocardium.

### 3.5. Actin Polymerisation in Integrin-Mediated Regulation of hiPSC-CM Ca^2+^ Cycling

We hypothesised that changes in the integrin-linked cytoskeleton mediated the changes in hiPSC-CM Ca^2+^ cycling properties. Blebbistatin-mediated inhibition of mechanical activation attenuated GRGDS-induced reduction in the Ca^2+^ transient TTP. However, the lack of effect of blebbistatin on T80 and the finding that modulation of Ca^2+^ transients is influenced by the sequence of amino acids flanking the RGD-motif suggest involvement of other mechanisms in a multifactorial signalling cascade.

### 3.6. Changes in CRU Ultrastructure

A key determinant of SR Ca^2+^ release is the apposition of the SR to the sarcolemma, which forms the CRUs [41,42]. CRUs contain two proteins essential to EC-coupling: LTCC and RyR2. This coupling has been demonstrated in both human atrial and human ventricular CMs [43]. We identified a greater co-localisation between LTCC and RyR2 following integrin ligand treatment.

We also investigated the co-localisation within CRUs—the size of the junctional gap between the SR and sarcolemma membrane. This space has previously been reported to be enriched with proteins that are recruited during formation of the CRU. The co-localisation is thought to be restricted by the size of the RyR2 cytoplasmic domain or foot [44], which assembles during CRU formation. Previous studies have identified the junctional gap to be between the range of 9–12 nm, with smaller junctional gaps being associated with fast contraction muscle fibres and wider junctions being found in slow twitch muscle fibres [45]. In our study, integrin ligand treatment increased the junctional gap distance compared to control (8.44 ± 0.273 nm vs. 6.65 ± 0.415 nm respectively, *p* = 0.0004, Appendix A) and to a junctional gap distance closer to in healthy adult CMs (9–12 nm [45]). The variability found within our study may reflect both the biological variability in samples but may also demonstrate the contrast in gap junction distance between fast and slow contraction muscle fibres, of which both are likely to be found within a hiPSC-CM population. We identified no change in the relative area of SR junctions with sarcolemma, in contrast to an earlier study which identified an increase in the area prior to birth before the T-system develops in rabbit CMs [46]. As the molecular machinery of CRUs is assembled in a series of steps, future studies should consider the incorporation of integrin ligands into the culture environment to support the development of CRUs. 

### 3.7. Integrin-Mediated Regulation of Cell Morphology and Sarcomere Length

In our study, we show that together with the improvement in Ca^2+^ cycling, integrin ligand–receptor interactions drive key changes in hiPSC-CM morphology. By labelling α-actinin (Figure 3D,E), we identified an increase in sarcomere length following ff_RGD treatment vs. control (Figure 3F; 1.75 ± 0.0309 μm vs. 1.63 ± 0.0201 μm, *p* = 0.0039). This is closer to the 2.0–2.2 μm widely considered to be the sarcomere length in the adult CM under load [47,48].

The average size of a human ventricular CM is 100–150 by 20–35 μm, significantly larger than neonatal or hiPSC-CMs. hiPSC-CMs show circular, triangular, or multi-angle morphology, unlike the elongated anisotropic shape that is typical of adult human CMs. During development, interactions with the ECM are crucial for cardiac morphogenesis [49]; at a single-cell level, collagen and integrin interactions promote a rod-shaped CM phenotype [50]. Our study identified that, although integrin ligand treatment causes morphological changes at the subcellular level closer to the adult CM, there was a reduction in hiPSC-CM cross-sectional area and volume following treatment with ff_RGD (Appendix A). We observed an increase in cellular aspect ratio, better resembling that reported in the adult CM, however still significantly short of this range (7–9.5 times [51]). We postulate that further medium-to-long term mechanical, electrical, or biochemical triggers during development and in disease drive CM hypertrophy and further increases in CM aspect ratio.

Our study presents important observations centred on the role that integrins play in modulating Ca^2+^ cycling in CMs. Integrins are a complex family of receptors important in a myriad of cellular processes. Future studies should investigate the integrins that play a significant role in modulation of Ca^2+^ cycling and the genomic changes that occur downstream of integrin ligand–receptor interactions. In doing so, we can improve our understanding of the CM maturation process during development and the physiological changes that occur in cardiac disease. 

## 4. Methods and Materials

### 4.1. Preparation and Maintenance of iCell CMs

Commercially available hiPSC-CMs (iCell Cardiomyocytes—Cellular Dynamics International, CA, USA) were resuspended and plated according to the manufacturer’s instructions. An experiment was a single biological replicate formed from a single batch of hiPSC-CMs, with the technical replicates indicated in the figure legends.

Cell resuspension occurred after 1 mL of a CM suspension was thawed, and the cellular content was calculated based on manufacturer-provided plating efficiency and viability measurements. Cells were re-suspended in commercially available plating medium (Cellular Dynamics International, Madison, WI, USA). All procedures were carried out under sterile conditions in a BioMAT 2 class II microbial safety cabinet. Cells were maintained in an incubator (Sanyo, MCO-5M, Japan) at 37 °C and 5% CO_2_.

HiPSC-CMs were plated onto 35 mm dishes with a 14 mm glass-bottomed well in the centre (MatTek Corporation, Ashland, MA, USA). Wells were pre-coated with a fibronectin solution for 1 h at 37 °C unless otherwise specified. Sixty thousand cells were used to achieve a confluent monolayer. Cells could adhere for two days, incubated at 37 °C and 5% CO_2_, before washing with phosphate-buffered saline (PBS) and changed over to maintenance medium (Cellular Dynamics International, Madison, WI, USA). Culture medium was replaced every two days, and the cells were used in experiments between days 10 and 14 after being thawed—at day 40–46 of differentiation.

### 4.2. Ca^2+^ Transient Measurements

Firstly, hiPSC-CMs were loaded with fluo4-AM (5 µM) (Thermo Fisher Scientific, MA, USA) at 37 °C for 20 min. Fluo-4 was excited using a 470 nm wavelength light-emitting diode (LED) and emitted fluorescence collected through a 530 ± 35 nm long-pass filter. All Ca^2+^ transient recordings were carried out in standard Tyrode’s solution (140 mM NaCl, 4.5 mM KCl, 10 mM glucose, 10 mM HEPES, 1 mM MgCl_2_, 1 mM CaCl_2_; pH 7.4) at 37 °C. The cells were field stimulated at 1 Hz using a 20 ± 10 V pulse of 5 ms duration, and recordings were captured using a NeuroCMOS camera (Redshirt Imaging, Decatur, GA, USA) at 0.5 kHz with a temporal bin of 2 (final frame rate 250/s) using a 40x oil-immersion objective. Recordings were made after the cells had reached steady state under field stimulation. The first four transients of each recording were signal-averaged before the following parameters were calculated using custom in-house software for MATLAB R2019b (MathWorks, Cambridge, UK): normalised fluorescent amplitude (F/F_0_, where F is peak fluorescence intensity and F_0_ is baseline fluorescence), time to peak (calculated as time from stimulus to peak fluorescence), and time to 80% decay (calculated as time from peak fluorescence to 80% reduction in amplitude). 

### 4.3. Assessment of Ca^2+^ Removal Mechanisms and SR Ca^2+^ Content

High concentrations of caffeine were locally applied to hiPSC-CMs to assess SR Ca^2+^ content and Ca^2+^ removal mechanisms. A solution of 40 mM caffeine in standard Tyrode’s was loaded into borosilicate glass micropipettes mounted on a 3-axis micromanipulator. The pipette was lowered into the bath, and the tip positioned above the cells at the far end of the field of view, upstream of the flow of bath solution. The cells were paced at 1 Hz before stimulation was ceased, and caffeine was applied simultaneously by applying positive pressure with a syringe pump, triggering a Ca^2+^ transient approximately 1 s following the last field stimulated Ca^2+^ transient. Caffeine flow was maintained until the end of the recording (Appendix A). To assess slow Ca^2+^ extrusion mechanisms, bath solution was switched to a Na^+^/Ca^2+^-free Tyrode’s solution (140 mM LiCl, 4.5 mM KCl, 10 mM glucose, 10 mM HEPES, 1 mM MgCl_2_, 1 mM EGTA; pH 7.4), and caffeine was applied as described above and elsewhere [17,35].

Ca^2+^ removal mechanisms were assessed by fitting a monoexponential curve to the declining phase of the Ca^2+^ transient. The rate of decline (k) was calculated as 1/tau (s^−1^). As set out by Bassani et al., the rate of decline of the field-stimulated transient (K_twitch_) is the sum of K_SR_, K_NCX_, and K_slow_ [52]. As such, the difference between K_twitch_ and the decline phase of the caffeine elicited Ca^2+^ transient was taken as k_SR_. The difference between the caffeine-elicited Ca^2+^ transients elicited in standard Tyrode’s solution and Na^+^/Ca^2+^-free Tyrode’s solution is attributed to k_NCX_. K_slow_ is equal to that of the decline phase of the caffeine-elicited Ca^2+^ transient in Na^+^/Ca^2+^-free Tyrode’s solution.

SR Ca^2+^ content measured as the caffeine induced Ca^2+^ transient amplitude. Fractional release of SR Ca^2+^ stores was measured as twitch Ca^2+^ transient amplitude divided by caffeine induced Ca^2+^ transient amplitude × 100.

### 4.4. Microelectrode AP Measurement

APs were recorded from a confluent hiPSC-CM monolayer using sharp borosilicate glass microelectrodes (30-0058, Harvard Apparatus, MA, USA). Pipettes with 30–60 M Ω resistance were prepared using a Sutter Instruments P-97 Pipette puller. Pipettes were filled with 2 M KCl, 0.1 mM EGTA, 5 mM HEPES at pH = 7.2. bath solution was Tyrode’s (1.8 mM CaCl, 140 mM NaCl, 4.5 KCl). Spontaneous action potentials were recorded in current clamp using a MultiClamp 700B amplifier (Molecular Devices, San Jose, CA, USA). 

### 4.5. AP Analysis

Aps were recorded from a confluent hiPSC-CM monolayer using sharp borosilicate glass microelectrodes (30-0058, Harvard Apparatus, MA, USA). Pipettes with 30–60 M Ω resistance were prepared using a Sutter Instruments P-97 Pipette puller. Pipettes were filled with 2 M KCl, 0.1 mM EGTA, 5 mM HEPES at pH = 7.2. bath solution was Tyrode’s (1.8 mM CaCl, 140 mM NaCl, 4.5 KCl). Spontaneous action potentials were recorded in current clamp using a MultiClamp 700B amplifier (Molecular Devices, San Jose, CA, USA).

### 4.6. ECM Substrate Preparation

hiPSC-CMs were prepared and seeded in monolayers as described above on dishes coated with four different ECM derivatives: gelatin (Sigma-Aldrich, St. Louis, MO, USA), fibronectin (Sigma-Aldrich, St. Louis, MO, USA), Matrigel (Corning Inc., Corning, NY, USA), and MAPTrix (mussel adhesive protein-based matrix; AMSBio, Oxford, UK). A 0.1% gelatin solution was prepared from a 2% stock solution by diluting in dH_2_O. Fibronectin solution was diluted to a final working concentration of 5 µg/mL in sterile PBS. Matrigel was used at a final concentration of 300 µg/mL. MapTrix was used at a final concentration of 5 µg/mL. Dishes were coated with all ECM substrates by incubating at 37 °C for at least two hours before cell seeding.

### 4.7. Soluble Integrin Stimulation Model

The RGD-containing peptide used was GRGDS (Gly-Arg-Gly-Asp-Ser) (Sigma-Aldrich, St. Louis, MO, USA). Peptide powder was made up to a stock solution of 50 mM in sterile dH_2_O. HiPSC-CM medium was removed, and the cells were washed with sterile PBS before medium with GRGDS (2 mM) was applied and compared to cultures without GRGDS as control. The cells were incubated for 24 h at 37 °C before electrophysiological assessment.

A type I collagen gel of 3 mg/mL was made from bovine collagen I (Thermo Fisher Scientific, Waltham, MA, USA) according to the manufacturer’s instructions. A 5 mg/mL stock solution of collagen was diluted in a combination of sterile 10X PBS, sterile dH_2_O, and 1 N NaOH. All solutions were kept on ice to prevent premature polymerisation. Gel polymerisation was activated by neutralising the collagen solution with 1 N NaOH to achieve an optimal pH of 7.0, assessed using universal indicator pH strips (pH 5.5–9.0; CamLab, Over, UK). For a GRGDS-collagen solution, GRGDS peptide in dH_2_O was used in place of dH_2_O alone.

For the application of the gel, hiPSC-CMs were washed in sterile PBS as before, and suction was used to clear the dish of as much liquid as possible. Approximately 100 µL of gel solution or phosphate-buffered saline control was pipetted onto the cells to cover the recessed glass coverslip in the dish. The cells and gel or control solution were incubated at 37 °C for 1 h to allow for gel polymerisation and attachment. After this time, serum-free medium was added and incubated again at 37 °C for 24 h before electrophysiological assessment.

### 4.8. Self-Assembling Integrin Ligands

The fibril-forming synthetic peptide studied was NH_3_^+^-RGDSGAITIGC-CONH_2_ (ff_RGD) (Genecust, Boynes, France). The synthetic lyophilised peptide powders were dissolved in serum-free media (M199, 1X ITS) at a concentration of 2 mM. The gel or control were mixed by vortexing and sonicated for 1 min. Gel or control were then incubated at 37 °C for 3 h to allow for self-assembly before 80 μL of the ff_RGD-containing gel or serum-free control was added above the hiPSC-CM monolayer and incubated for 24 h.

### 4.9. LIVE/DEAD Viability/Cytotoxicity Assay

The LIVE/DEAD viability/cytotoxicity assay was used to detect the % of living cells in ff_RGD-treated and control CM monolayers. A stock solution of 2 µM calcein AM and 4 µM ethidium homodimer-1 was prepared. A total of 20 µL of 2 mM ethidium homodimer-1 and 5 µL of 4 mM calcein AM were added to 10 mL of serum free medium. The stock solution was then vortexed to ensure thorough mixing. Glass-bottomed dishes with spontaneously beating hiPSC-CM cultures were treated with ff_RGD (2 mM) or serum free for 24 h control was filled with 200 µL of stock solution and incubated at 37 °C in a 5% CO_2_ incubator for 20 min. In order to assess the fluorescence of the dishes, images were collected using a 10× objective on a Zeiss AxioObserver Confocal microscope equipped with a motorised stage. Analysis of the images was conducted offline using Fiji software.

### 4.10. Electron Microscopy

HiPSC-CMs were attached to 1 × 1 cm Aclar film squares treated with 10 µg/mL Matrigel in serum-free medium and cultured in maintenance medium. Self-assembling integrin ligand ff_RGD or control was formed as described above and 2 mM of the gel or control was added above the hiPSC-CM monolayer and incubated for 24 h. Samples were washed in PBS before fixing in 2.5% glutaraldehyde and 2% paraformaldehyde in 100 mM sodium cacodylate buffer. After washing in sodium cacodylate buffer, samples were subsequently stained in 1% osmium tetroxide and 1.5% potassium ferrocyanide, 1% tannic acid, 1% osmium tetroxide and 1% uranyl acetate, with washings in water after each staining step. After staining, samples were dehydrated in an ethanol ascending series (50%, 70%, 90%, 100%, 100%) followed by further dehydration in pure acetone. Increasing concentrations of TAAB 812 hard resin (25%, 50%, 75%, 100%) mixed with acetone were used for infiltration. To finish, samples were embedded in pure resin and cured at 60 °C for 36 h.

Resin blocks were sectioned with a diamond knife at 80 nm on a Leica Ultracut UCT, and sections were imaged on an FEI Tecnai12 BioTwin operated at 80 kV. Images were collected in dm3 format (Gatan Digital Micrograph) and analysed in ImageJ or IMOD [53,54,55].

### 4.11. Confocal Image Acquisition

Confocal microscopy was carried out at the Facility for Imaging by Light Microscopy (Imperial College London, London, UK). A Zeiss LSM780 confocal laser scanning microscope with 7 laser lines (Diode 405 nm 30 mW, Argon multiline 458/488/514 nm 25 mW, HeNe 543 nm 1 mW, HeNe 594 2 mW and HeNe 633 nm 5 mW) and 34 detectors (34 Channel GaAsP Detection System) was used for all imaging. During all experiments, laser intensity was optimised to enhance resolution but minimise photobleaching and saturation. Z-stacks of images were acquired if cell volume was required for analysis.

### 4.12. Analysis of Sarcomere Length

The average sarcomere length was measured in ImageJ by separating the α-actinin channel and viewing the profile plot of a line drawn perpendicularly across the α-actinin bands. Each peak on the graph corresponds to the increased fluorescence caused by an α-actinin band; thus, the distances between the 5 peaks were measured and averaged to give sarcomere length [56].

### 4.13. Analysis of Cell Area

Analysis of the average CM area was performed using WGA-stained images acquired using a 20× objective. A region of interest was drawn around the border of individual CMs, and the area was then measured digitally using ImageJ. The areas of 4 CMs were measured per dish. Three dishes were analysed per condition.

### 4.14. Analysis of Cell Aspect Ratio and Cell Volume

WGA-stained images were used to calculate the aspect ratio of the cell and nucleus. The major and minor axes were measured digitally using ImageJ, and the aspect ratio was calculated as the major axis divided by the minor axis. Z-stacks of hiPSC-CMs stained with WGA were used to calculate cell volume. Individual images were acquired at 0.73 µm intervals. The areas of the lowest and highest stack images were measured, as well as the middle image in the series. These areas were averaged and multiplied by the stack height to give the cell volume.

### 4.15. Bright Field Imaging

Brightfield images of hiPSC-CMs were obtained using a 40× objective mounted on a Zeiss Axio Observer Inverted Widefield Microscope at 20 frames/s. The samples were kept in 37 °C standard Tyrode’s. All the acquired images were processed in ImageJ (NIH).

### 4.16. Immunostaining

Cells were plated either on glass-bottomed dishes as described or on glass coverslips and maintained for 10 days. The culture medium was then removed, and the cells were washed in PBS (Sigma-Aldrich, St. Louis, MO, USA) before fixing in 4% PFA for 20 min. Preparations were washed in PBS three times for 5 min each and used immediately or stored at 4 °C in PBS. PFA-fixed preparations were subsequently permeabilised in 0.2% Triton X-100 (Sigma-Aldrich, St. Louis, MO, USA) in PBS for 3 min followed by three 5 min washes in PBS.

Preparations were incubated in blocking solution containing 3% BSA (Sigma-Aldrich, St. Louis, MO, USA) in PBS for 1 h at room temperature. The primary antibodies mouse IgG1 anti-α sarcomeric actinin were diluted 1/500 in PBS. 

In the co-localisation immunostaining, preparations were stained with mouse monoclonal antibody IgG1 anti-RyR2 in a 1/500 dilution in PBS and guinea pig polyclonal antibody (Alomone Labs, Jerusalem, Israel) anti-1.2 CaV in 1/250 dilution in PBS.

Cells were washed three times in PBS and incubated with secondary antibodies for 1.5 h at room temperature. Cells were then washed three times and mounted with a DAPI-containing mounting solution (Vectashield, Vector Labs, Burlingame, CA, USA).

### 4.17. WGA Staining

WGA staining was carried out to label the cell membranes. Cells on dishes were washed three times in PBS for 5 min each time and then fixed with 3.7% PFA for 15 min. The samples were washed three times for 5 min each to ensure that any remaining fixative was removed. According to the manufacturer’s protocol, the samples were incubated with 5 μg/mL WGA solution (L4895, Sigma-Aldrich, St. Louis, MO, USA) for 10 min at room temperature (Invitrogen, 2009; Wright, 1984). After the specimens were washed in PBS, the nuclei were counterstained with Vectashield mounting medium containing Hoechst, and the samples were mounted and kept refrigerated without light exposure until they were imaged using an inverted Zeiss LSM-780 confocal microscope.

### 4.18. Co-Localisation and Quantification Analysis 

To measure co-localisation between LTCCs and RyRs on ImageJ, the merged images from the monolayers were split in separate channels, and brightness/contrast was adjusted to ‘Auto’ for each channel then merged again to create a Z-project. Using the Z-stack composite, the borders from individual cells were identified using the WGA membrane staining. The option ‘clear outside’ was selected to isolate individual cells from the preparations. Once channels were split again, the ones from RyRs and DHPRs were inputted in JACoP plugin for co-localisation on ImageJ [57]. Pearson’s coefficient and Mander’s coefficients were used for the analysis. A threshold was set manually to measure the pixel intensity for each image and kept constant for the analysis of all the following images. Two Mander’s coefficients were analysed for each cell—LTCCs overlapping a RyR background (M1) and RyRs overlapping a LTCC background (M2).

### 4.19. Statistical Analysis

Cell parameter data were calculated manually and blindly by two operators not involved in the data analysis. Before statistical tests, all data were subjected to the D’Agostino and Pearson omnibus normality test. For parametric distribution, a two-way paired *t*-test was used. For comparisons of >2 sample groups, a one-way within-subject ANOVA (analysis of variance) was used. Post hoc tests used with ANOVA were either Tukey’s test (for comparison of all means) or Dunnett’s test (for comparison to a single control value). For non-parametric distribution, a statistical test was performed using Mann–Whitney (2 sample groups) or Kruskal–Wallis (>2 sample groups). The data are expressed as means *±* standard error of mean (SEM). * = *p* < 0.05, ** = *p* < 0.01, *** = *p* < 0.001, **** = *p* < 0.0001.

## 5. Conclusions

The role of extracellular signals involved in the modulation of EC-coupling is multifaceted. This study demonstrates that integrin-binding peptides improve hiPSC-CM function through increased functional recruitment of the SR with improved CICR, and alterations in CM ultrastructure, increasing co-localisation between L-type Ca^2+^ channels and ryanodine receptors and improving cell alignment. These findings are critical in understanding the mechanisms involved in myocardial plasticity and demonstrate the functional importance of the ECM in modulating CM function in physiology.

## Figures and Tables

**Figure 1 ijms-23-10940-f001:**
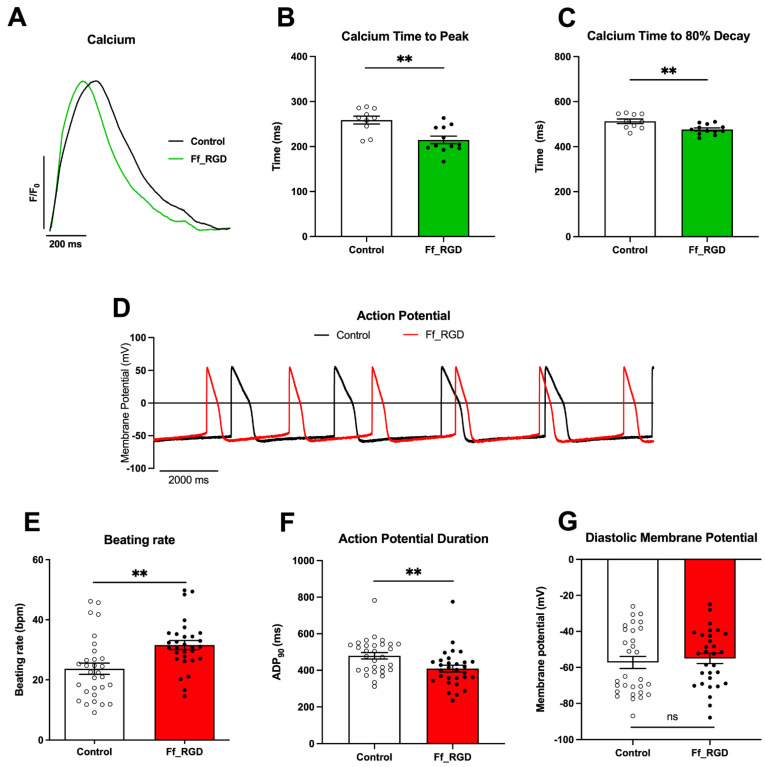
Fibril−forming integrin ligands module cardiomyocyte Ca^2+^ transients and action potential morphology. (**A**) Ca^2+^ transient representative traces after 24 h treatment with ff_RGD (2 mM) (green) compared to control (black) during 1 Hz field−stimulation. Ca^2+^ transient (**B**) TTP (** *p* = 0.0016) and (**C**) T80 illustrated as means ± SEM (** *p* = 0.0055) (*n* = 10 control, *n* = 12 ff_RGD images from four batches). (**D**) AP representative traces of spontaneously beating ff_RGD−treated cardiomyocytes (red) compared to control (black). Parameters measured were (**E**) beating rate (** *p* = 0.0018), (**F**) AP duration (** *p* = 0.0029), and (**G**) maximum diastolic membrane potential (ns *p* = 0.602) illustrated as means ± SEM (*n* = 30 control, *n* = 30 ff_RGD cells from 10 dishes, duplicates from five batches). hiPSC−CMs were used at day 40–46 of differentiation.

**Figure 2 ijms-23-10940-f002:**
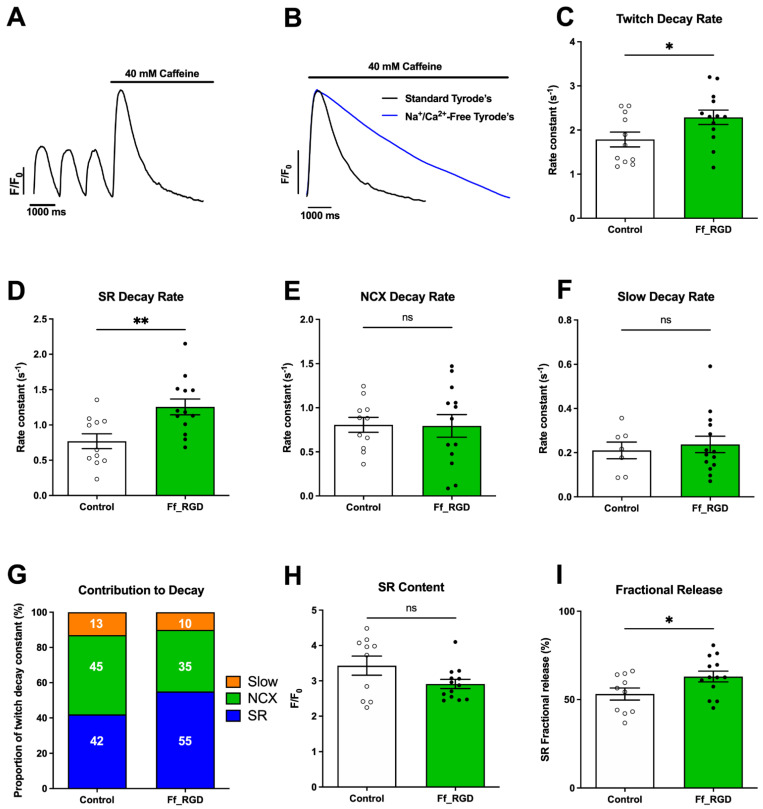
Integrin ligands recruit the sarcoplasmic reticulum (SR) for Ca^2+^ cycling at 1 Hz field−stimulation. (**A**) Representative trace of three 1 Hz field−stimulated (twitch) Ca^2+^ transients followed by caffeine response (see Appendix A). (**B**) Representative traces showing response to caffeine in Na^+^/Ca^2+^-free and standard Tyrode’s solution. Data illustrated as means ± SEM. Parameters were rate of (**C**) twitch Ca^2+^ transient decay (* *p* = 0.0446), (**D**) SR Ca^2+^ uptake (** *p* = 0.0049), (**E**) sodium/calcium exchanger (NCX)−mediated Ca^2+^ removal (ns *p* = 0.941), (**F**) Ca^2+^ removal mediated via mitochondrial Ca^2+^−uniporter and sarcolemma Ca^2+^−ATPase activity (slow) (ns *p* = 0.856). Rates of removal mechanisms were used to calculate (**G**) percentage contribution of Ca^2+^ removal mechanisms. (**H**) SR Ca^2+^ content (ns *p* = 0.376). (**I**) Fractional release of SR Ca^2+^ stores (* *p* = 0.0426) (Panel **C**–**E**,**H**,**I**—*n* = 11 control, *n* = 13 ff_RGD images with averaged signal from within the camera field of view from at least four batches. Panel F *n* = 10 control, *n* = 14 ff_RGD images with averaged signal from within the camera field of view from at least four batches). Cardiomyocytes were used at day 40–46 of differentiation.

**Figure 3 ijms-23-10940-f003:**
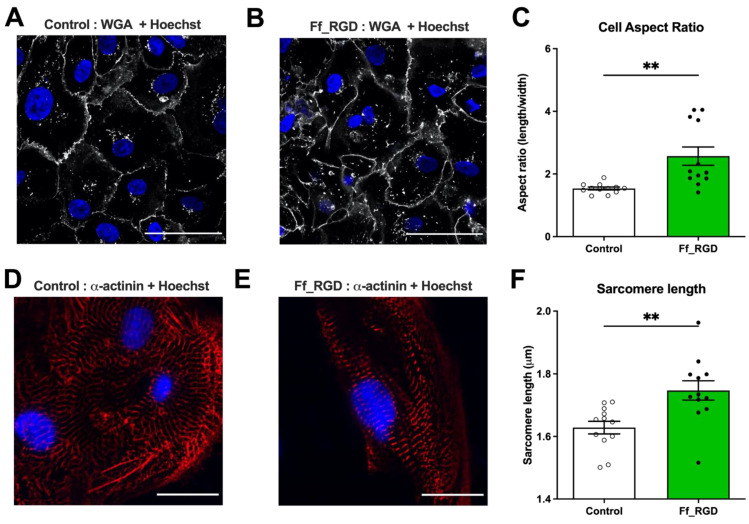
Integrin ligand ff_RGD induces changes in cardiomyocyte structure. Cardiomyocytes at day 40–46 of differentiation were treated for 24 h with (**A**) control or (**B**) ff_RGD. Representative confocal images with wheat germ agglutinin (WGA) (white) and Hoechst (blue) staining. Scale bar = 50 μm. Confocal images were used to investigate the (**C**) aspect ratio (** *p* = 0.0021) illustrated as means ± SEM (*n* = 12 control, *n* = 12 ff_RGD images from three batches. Each data point is an average of the cellular values within the field of view. Hoechst (blue) and α−sarcomeric actinin (red) staining of (**D**) control and (**E**) ff_RGD−treated cardiomyocytes. Scale bar = 50 μm. α−sarcomeric actinin used to measure (**F**) sarcomere length (** *p* = 0.0039) illustrated as means ± SEM (*n* = 12 control, *n* = 12 ff_RGD cells from three batches. Each data point is an average length of 20 sarcomeres from a cell).

**Figure 4 ijms-23-10940-f004:**
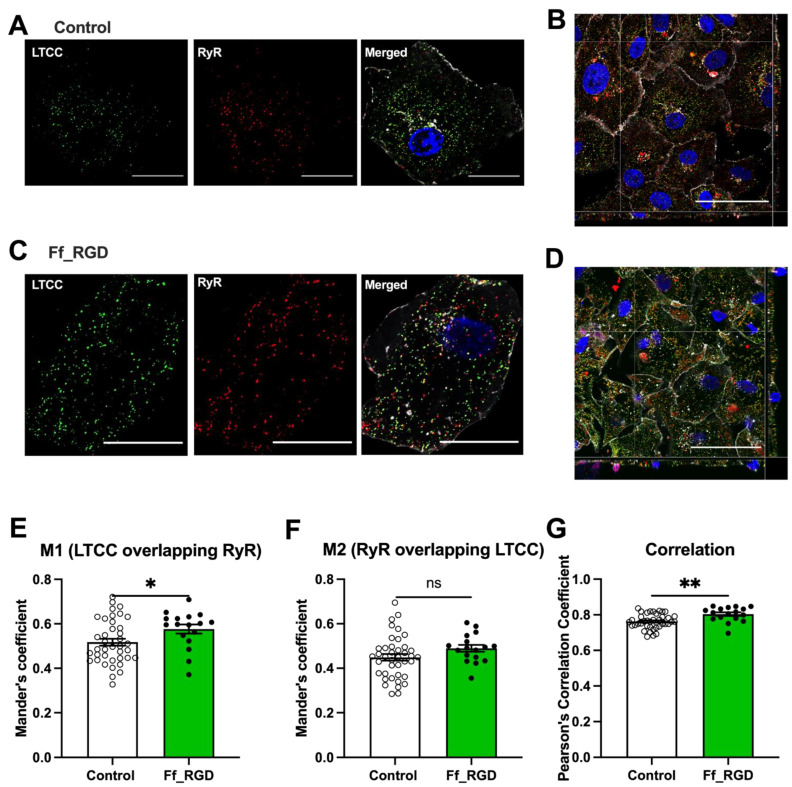
Integrin ligand ff_RGD induced changes in the Ca^2+^ cycling ultrastructure. L-type calcium channel (LTCC) subunit CaV1.2 (green) and ryanodine receptor (RyR) 2 (red) were labelled and orthoview z-stack formed with wheat germ agglutinin (WGA) (white) and Hoechst (blue) staining (**A**,**B**) control and (**C**,**D**) ff_RGD-treated cardiomyocytes (day 40–46 of differentiation). Scale bar for A and 6 = 20 μm, B and D = 50 μm. Merged images used to calculate the Mander’s correlation coefficient for (**E**) LTCC overlapping RyR background pixels (M1) (* *p* = 0.0363), (**F**) RyR overlapping LTCC background pixels (M2) (ns *p* = 0.1143), and (**G**) Pearson’s correlation coefficient between LTCC and RyR (** *p* = 0.001) illustrated as means ± SEM (*n* = 39 control, *n* = 18 ff_RGD cells from ten batches).

## Data Availability

Data is available upon reasonable request to the corresponding authors.

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
