# Peer review of "Integrins Increase Sarcoplasmic Reticulum Activity for Excitation—Contraction Coupling in Human Stem Cell-Derived Cardiomyocytes"

_ijms, 2022, doi:10.3390/ijms231810940_

Round 1

Reviewer 1 Report

The study by Wang et al. reports the results of studies using a RGD fibrils to  probe the relationship between integrin and SR activity. The study is well written and the overall conclusions are reasonable. After reviewing the manuscript, I have a few concerns/questions that I think should be addressed in the manuscript.

1. A better description of the surface chemistry would be helpful

2. A satisfying explanation for the sensitivity of each flanking amino acid of the RGD sequence is not provided. Could the authors provide more information?

3. In Figure 3F the authors plot the cell length and conclude that ff_RGD cells have a greater length than control. In terms of length, is this increase comparable to cell length from in vivo tissue?

4. Why is there so much spread in the data for the values for Supplementary Figure 3C?

5. The authors conclude that the ff_RGD aids in altering the Ca2+ cycling maturity. The authors should check downstream effects on the cell by q-PCR to determine if the Ca2+ cycling maturity physiologicaly impacts the transcription of downstreram genes and whether the cell has changed.

Reviewer 2 Report

The manuscript by Wang et al. describes the influence of RGD peptides and RGD-containing ECM proteins over EC coupling and increase Ca++ cycling rate mediated by sarcoplasmisc reticulum in hiPSC-derived cardiomyocytes.

The experimental design is interesting and provides new insights on the role of ECM proteins in CM physiology, nevertheless some issues should be addressed before drawing firm conclusions:

1) integrins are a large family of  dimeric receptors composed of  24 different members, formed by 18 alpha and 8 beta subunits in combination. Of these receptors, only a sub-family binds RGD-containing peptides (Ludwig et al, cancers 2021). Based on this, a molecular characterization of integrin expression in hiPSC-derived CM by qPCR is required. In alternative, evaluation of integrin expression on the cell surface by FACS analysis using specific antibodies should be performed but, in this case, a precise knowledge of integrin expression pattern is necessary.

2) The Authors should demonstrate that the peptides used in their work actually activate RGD_binding integrins by investigating downstream effectors activation (eg. FAK phosphorylation).

3) In addition to scrambled peptides or RGD analogs, RGD antagonists like cilengitide or RGD-binding integrins neutralizing antibodies should be used.
